# Narrative Style and the Spread of Health Misinformation on Twitter

**Achyutarama R. Ganti[1]\***, **Eslam Hussein[2]\***,
**Steven R. Wilson[1]**, **Zexin Ma[3]**, **Xinyan Zhao[4]**
[1]Oakland University, [2]Virginia Tech,
[3] University of Connecticut, [4] University of North Carolina
ganti@oakland.edu, ehussein@vt.edu,
stevenwilson@oakland.edu, zexin.ma@uconn.edu, ezhao@unc.edu

## Abstract

Using a narrative style is an effective way to communicate health information both on and off social media. Given the amount of misinformation being spread online and its potential negative effects, it is crucial to investigate the interplay between narrative communication style and misinformative health content on user engagement on social media platforms. To explore this in the context of Twitter, we start with previously annotated health misinformation tweets ($n \approx 15,000$) and annotate a subset of the data ($n = 3,000$) for the presence of narrative style. We then use these manually assigned labels to train text classifiers, experimenting with supervised fine-tuning and in-context learning for automatic narrative detection. We use our best model to label remaining portion of the dataset, then statistically analyze the relationship between narrative style, misinformation, and user-level features on engagement, finding that narrative use is connected to increased tweet engagement and can, in some cases, lead to increased engagement with misinformation. Finally, we analyze the general categories of language used in narratives and health misinformation in our dataset.

## 1 Introduction

A narrative is a fundamental form of human communication. Although colloquially used interchangeably with "story", the formal concept of narrative is defined more broadly—consisting of both story (connected events and characters) and narrative discourse (how the story is told) (Abbott, 2008). In other words, a narrative represents the presentation of a sequence of events experienced by a character or characters (Bilandzic and Busselle, 2012a; Dahlstrom, 2021). This study uses this definition to label and classify tweets into narrative and non-narrative based on whether or not they are written in a narrative style. An example

narrative tweet is, "We got our first dose of the vaccine! Can't tell you how excited and thankful we feel, and can't wait to get the next dose".

Considering the power of narrative information in influencing individuals' health-related beliefs, attitudes, and behaviors (Ma and Yang, 2022; Murphy et al., 2013), it is important to understand and address narrative *misinformation* on social media. In particular, research has found that online health misinformation is often created using narratives (Peng et al., 2022). Further, emerging research indicates that individuals are less likely to discern or verify misinformation when it is presented in a narrative (vs. non-narrative) format, possibly because narrative misinformation is perceived as more relatable and engaging (Zhao and Tsang, 2023). Therefore, it is crucial to classify misinformation as narrative and non-narrative and examine how this affects user engagement on social media.

Misinformation is defined as false or misleading information that is contrary to the consensus of the scientific community based on the best available evidence at the time (Vraga and Bode, 2020). The prevalence and diffusion of incorrect information and fabricated narratives on social media have been a growing concern (Southwell et al., 2022). According to one study, 28% of the most viewed coronavirus videos included non-factual information, totaling 62 million views as of March 21, 2020 (Li et al., 2020). US-based respondents of another survey estimate that about 33% of social media content that they view contains some misinformation (Nielsen et al., 2020).

How individuals assess and act on misinformation can be affected by many factors, such as message features, source credibility, and individual motivations and emotions (Marsh and Yang, 2018). Existing research has shown that individuals are particularly susceptible to narrative misinformation that uses negative emotions such as anger or fear, incorrect personal stories, and made-up testimonials

---

*Equal contribution, listed in alphabetical order.

(Zhao and Tsang, 2023; Lee and Shin, 2022). Yet, most studies were conducted in a laboratory setting, limiting their ecological validity and generalizability. Further, prior studies have not quantitatively examined how social media misinformation in the narrative (vs. non-narrative) form affects user engagement at scale. Therefore, in this study, we use a large set of health tweets to investigate the presence of narrative misinformation and its impact on user engagement, and explore the features of both narratives and misinformation on Twitter.

The contributions[1] of this work are (1) narrativity annotations for two existing misinformation datasets, which can be used to study the interaction between narrative and misinformation on Twitter; (2) experiments with several types of models for narrative detection, and the use of the best model to label all of the available data; and (3) analyses of the relationship between narrative style and user engagement in the presence of misinformation, as well as variations in the categories of language used in both narrative and non-narrative tweets. We find that tweets containing narratives typically had higher engagement, while the presence of misinformation was associated with less engagement. Especially in the context of vaccine-related content, using narratives was related to increased engagement with misinformation, even while controlling for number of followers. Additionally, through our analyses, we find supporting evidence for existing studies on the linguistic features of narratives even in the social media context.

## 2   Related Work

Prior work established the importance of narratives both on social media and in the context of sharing health-related information. On Twitter, specifically, Dayter (2015) analyzed multi-tweet stories to study the importance of storytelling for identity construction. In the health domain, Overcash (2003) emphasized the importance of narrative research in developing healthcare services for patients, while Andy et al. (2021) studied how users on online health communities show support for one another. Prior research suggests that narratives are more effective at communicating health risks than non-narratives (Janssen et al., 2013; Ma, 2021) and promoting health behavior change (Kreuter et al., 2010). Guidry et al. (2015) found that anti-vaccine

pins on pinterest used more narrative information while pro-vaccine used more statistical information, and that the latter type had higher levels of user engagement. Additionally, Betsch et al. (2011) studied the influence of narratives on the perception of vaccination risks, and highlighted the effectiveness of narrative communication in influencing human decision making about vaccine risk.

Issues with the spread of misinformation on platforms like Twitter are also widely researched. For example, Sharevski et al. (2022) studied COVID-19 misinformation on Twitter and the change in user perception of COVID-19 vaccine content as a result of soft moderation of misinformation content on the platform. The spread of misinformation on social media is not just limited to the common public: a study done by Shahi et al. (2021) reveals that verified Twitter handles(organizations/celebrities) are also responsible for creating and spreading misinformation. The authors also observed that fake claims propagate faster than semi-fake claims, and tweets containing misinformation often attempt to discredit other information on social media.

While most studies of narratives have relied on manual annotations, Dirkson et al. (2019) and Verberne et al. (2019) used human-assigned labels to train text classification models to detect narrativity in social media posts in the health domain. Both studies focused on bag-of-words models with n-gram features, while more recent research found that deep learning models were more successful than classical machine learning models at detecting narratives in Facebook posts (Ganti et al., 2022). These studies suggest that automatic narrative classification could enable the labeling of even larger social media datasets.

## 3   Data Collection and Annotation

We leveraged existing misinformation datasets and provided new annotations for the presence of narratives to create a single dataset that contained both sets of labels for each text: misinformation (or not) and narrative (or not).

### 3.1   Data Collection

We selected two datasets: *ANTiVax* (Hayawi et al., 2022), and *CMU-MisCov19* (Memon et al., 2020) based on three criteria: the quality of the annotation, relevance to the topic investigated, and the amount of data. Each dataset was annotated for medical misinformation by subject matter experts.

---

[1]Code and other resources related to this work available at: https://github.com/ou-nlp/NarrMisinfoEMNLP23

Table 1 shows the number of tweets collected by the authors of each dataset and the number of hydrated tweets that were available at the time of our data collection (September 2022), which is the set of tweets used in this work. Most of the unavailable tweets fall under the misinformation category.

The **ANTiVax** dataset was collected to train machine learning algorithms to classify and detect COVID-19 vaccine misinformation. Authors used the Twitter API to collect tweets related to the COVID-19 vaccines using keywords (e.g., vaccine, Pfizer, Moderna). Tweets were annotated into misinformation (**misinformed**) tweets and general vaccine-related tweets (**informed**), and annotations were verified by medical experts.

The **CMU-MisCov19** dataset was collected based on tweets of two groups: informed and misinformed users who wrote tweets related to COVID-19. Authors used the Twitter API to collect tweets related to COVID-19 using keywords (e.g., coronavirus, covid). Tweets were annotated into five false information categories and five true information categories, which we merge into one **misinformed** and one **informed** category, respectively.

## 3.2 Annotation Process

Our annotation guidelines were developed iteratively through discussion and revision. During each of our four pilot annotation phases, three of the authors annotated between 20-100 randomly sampled tweets for narrative style. After each round of annotation, the annotators met together along with experts in narrative communication to discuss the annotated tweets, and updated the annotation guidelines from the previous round. After four rounds, all authors reached a consensus on the final annotation guidelines (appendix A).

After finalizing the annotation guidelines, we sampled 3,000 tweets (20.6% of the 14,561 hydrated tweets in our dataset) stratified with respect to dataset and misinformation stance. Each tweet was initially independently annotated by two of the three annotators who developed the guidelines, with agreement measured using Krippendorff's al-

| Datasets | ANTiVax | | | CMU-MisCov19 | | | Combined | | |
|---|---|---|---|---|---|---|---|---|---|
| Info/Misinf | Info | Misinf | Total | Info | Misinf | Total | Info | Misinf | Total |
| Collected | 9322 | 5751 | 15073 | 2146 | 1423 | 3569 | 11468 | 7174 | 18642 |
| Hydrated | 8059 | 3939 | 11998 | 1772 | 791 | 2563 | 9831 | 4730 | 14561 |
| % | 55.3% | 27% | 82.4% | 12.2% | 5.4% | 17.6% | 67.5% | 32.5% | 100% |
| % unavail. | 13.5% | 31.5% | 20.4% | 17.4% | 44.4% | 28.2% | 14.3% | 34.1% | 21.9% |

Table 1: Number of available tweets in both datasets after rehydration.

pha = 0.71. If the two annotators agreed, the agreed upon label was used, otherwise the third annotator stepped in to break the tie and the majority label was used in our final dataset.

## 4 Narrative Detection

Given the annotated tweets, we then set out to train classification models that could be used to label the entirety of the misinformation datasets that we collected.

### 4.1 Classification Methodology

The annotated data were shuffled and split 80-10-10 for training, validation, and testing. We explore three categories of models: classical machine learning models with bag-of-words features, fine-tuned transformer encoder models, and auto-regressive generative models with in-context learning. All models are trained three times with different random seeds, and the average results are presented.

The bag-of-words-based machine learning models we used were scikit-learn's (Pedregosa et al., 2011) implementations of Naive Bayes, Support Vector Machine classification, and Logistic Regression. We conducted hyperparameter tuning as outlined in Appendix B.

For the transformer-based deep-learning models, we considered BERT (Devlin et al., 2019), DistilBERT (Sanh et al., 2019), RoBERTa (Liu et al., 2019), CardifNLP's TwitterRoBERTa (Barbieri et al., 2020) and DeBERTa (He et al., 2021) models available on HuggingFace Wolf et al. (2019) using `bert-base-uncased`, `distilbert-base-uncased`, `roberta-base`, `cardiffnlp/twitter-roberta-base` and `microsoft/deberta-v3-base` checkpoints, each with their default tokenizers and the output of the `[CLS]` input token as the input to a trainable classification layer. (training hyperparameters listed in Appendix B).

For the generative models, we used the `gpt-3.5-turbo` and GPT-3 (Brown et al., 2020) `text-davinci-003` models from the OpenAI API [2]. Both models take in an instruction or a prompt as the input and respond with completion to match our context or question. The goal of this experiment was to investigate if auto-regressive models such as GPT-3 and GPT-3.5-Turbo are capable of detecting the presence of narratives in tweets with little or no training data, which, if very successful,

---
[2]https://openai.com/product

Figure 1: Prompt components for GPT Models. From top to bottom, the blocks display the **definition** (blue), **instructions** (orange), **few-shot** examples (green), and the target **tweet** to be classified (yellow).

|  | GPT-3 (Davinci) | | | GPT-3.5-Turbo | | |
|---|---|---|---|---|---|---|
| **Model** | **F1** | **Prec** | **Rec** | **F1** | **Prec** | **Rec** |
| zero-shot | 0.484 | 0.938 | 0.326 | 0.302 | 0.402 | 0.241 |
| +def | 0.183 | 0.939 | 0.101 | 0.080 | 0.485 | 0.040 |
| +instr | 0.328 | 0.801 | 0.206 | 0.240 | 0.400 | 0.171 |
| +def+instr | 0.489 | 0.864 | 0.341 | 0.466 | **0.767** | 0.335 |
| few-shot | 0.699 | 0.835 | **0.601** | 0.584 | 0.460 | **0.799** |
| +def | 0.697 | 0.879 | 0.578 | **0.675** | 0.598 | 0.780 |
| +instr | 0.698 | 0.941 | 0.555 | 0.642 | 0.554 | 0.763 |
| +def+instr | **0.712** | **0.952** | 0.570 | 0.636 | 0.546 | 0.762 |

Table 2: Narrative class F1, Precision and Recall, averaged across three runs, for GPT-3 models trained on detecting narratives using the validation set only. **Bold** indicates the top score for each metric per model.

| Model | F1 | Prec | Recall |
|---|---|---|---|
| Logistic Reg | 0.704 | 0.800 | 0.629 |
| GPT-3 | 0.718 | **0.977** | 0.568 |
| Naive Bayes | 0.734 | 0.708 | 0.761 |
| SVM | 0.775 | 0.680 | 0.900 |
| BERT | 0.886 | 0.873 | 0.900 |
| TwitterRoBERTa | 0.887 | 0.885 | 0.888 |
| DistilBERT | 0.893 | 0.900 | 0.886 |
| DeBERTa | 0.910 | 0.907 | 0.914 |
| RoBERTa | **0.924** | 0.918 | **0.931** |

Table 3: Narrative class F1, Precision, and Recall scores, *averaged across three runs* for the text classification models. The best score for each metric is in **bold**. GPT-3 indicates the result of the best GPT-3 model based on various prompting schemes.

could reduce or avoid the entire manual annotation process that was required to obtain our initial set of narrative labels.

The experimental setup for the auto-regressive models consists of two main settings: zero-shot and few-shot. In the zero-shot setting, the model is shown the target tweet and asked "Is this tweet a narrative?", while in the few-shot setting, several example tweets with their correct labels are prepended to the prompt in a question-answer format, with the labeled examples being evenly distributed across the two datasets and narrativity labels, i.e., $n$ narratives and $n$ non-narratives from each dataset are used, leading to $4n$ total examples. We tune the value of $n$ using the validation set and set $n = 5$ for our experiments. Within each setting (zero- or few-shot), we also experimented with the inclusion of the definitions of narratives and additional guidelines that the annotators developed and used during the annotation process. A summary of the components that can be included or removed from the prompt is outlined in Figure 1.

## 4.2 Experimental Results

Given that we explore eight configurations for the GPT models, we first use the validation set to select the best performing setup (Table 2). The F1 scores for both the GPT models across all the few-shot

experiments is reported in Appendix C. The GPT-3 based `text-davinci-003` model outperforms `gpt-3.5-turbo` in nearly all of the zero-shot and few-shot experiments. The best overall GPT model, based on `text-davinci-003` used few-shot learning and included the narrative definitions and labeling guidelines in the prompt, and we evaluate this model on the test set (GPT-3 in Table 3).

The results for all models on the test set are presented in Table 3. Our findings align with those of Ganti et al. (2022), who found that transformer-based models consistently outperform bag-of-words models for narrative detection, with the fine-tuned base RoBERTa model performing best overall, outperforming even the version of RoBERTa pretrained specifically on Twitter data. The GPT-3 `text-davinci-003` model achieved high precision but lacked in recall, resulting in an F1-score similar to the bag-of-words models, though it requires less than 1% of the amount of training data (in the form of examples for in-context learning). In the end, we use our top performing fine-tuned RoBERTa model to label our entire dataset, which we analyze in the next section.

| Datasets | ANTiVax | | | CMU-MisCov19 | | | Combined | | |
|---|---|---|---|---|---|---|---|---|---|
| Info or Misinfo | Info | Misinf | Total | Info | Misinf | Total | Info | Misinf | Total |
| Narrative Narr | 5077 | 492 | 5569 | 274 | 78 | 352 | 5351 | 570 | 5921 |
| Narrative Non | 2982 | 3447 | 6429 | 1498 | 713 | 2211 | 4480 | 4160 | 8640 |
| Total | 8059 | 3939 | 11998 | 1772 | 791 | 2563 | 9831 | 4730 | 14561 |

Table 4: Number of tweets in both datasets after applying the fine-tuned RoBERTa model on the unlabeled portion of the dataset.

## 5 Narrative and Misinformation

In this section, we investigate the effect of narrativity and misinformation on user engagement metrics and variations in linguistic features. First, we analyze the distribution of each category of tweets in the full set of $\sim 14.5k$ tweets labeled by our annotators and the best RoBERTa model from the previous section. Second, we discuss how the dimensions of narrativity and misinformation affect user engagement (i.e., counts of retweets and likes). Finally, we investigate how linguistic variations are significant between narratives and non-narratives and between informed and misinformed tweets.

### 5.1 Data Analysis

After applying the best model on the unlabeled portion of the dataset, we have the narrativity distribution in table 4. Where 46.4%, 13.7%, and 40.7% are narrative tweets of the ANTiVax, CMU-MisCov19, and combined datasets, respectively.

**Narrativity**. Although the ratio between informed and misinformed tweets in the ANTiVax, CMU-MisCov19 datasets are 2.07 and 2.24, respectively, we notice that narrativity is used more in informed than misinformed tweets, where informed narratives are 10.32, 3.51, and 9.39 times more common than misinformed narratives in the ANTiVax, CMU-MisCov19, and combined datasets, respectively. On the other hand, the non-narrative style is used more in misinformed tweets in ANTiVax and more in informed tweets in the CMU-MisCov19 and combined datasets. We can see that narrativity is a more commonly used communication style among informed users to share their vaccination and COVID-19 experiences than misinformed users. We also observe that narrativity is less prevalent in the CMU-MisCov19 than the ANTiVax dataset, with narrative to non-narrative ratios of 0.27 and 1.14, respectively.

**Misinformation**. Informed tweets are more prevalent in the ANTiVax and CMU-MisCov19 datasets by a ratio of 2.05 and 2.24, respectively, where narrative to non-narrative *informed* tweets

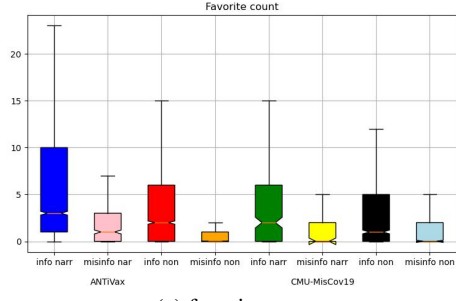

(a) favorite count

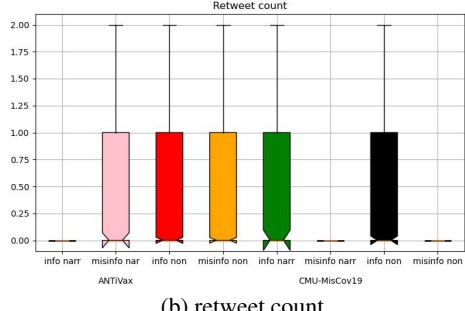

(b) retweet count

Figure 2: Box plots of (a) favorite count and (b) retweet count for each of the datasets under each misinformation and narrativity category. Note: outliers are removed.

have ratios of 1.70 and 0.18 in each dataset, respectively, while the ratio of narrative to non-narrative *misinformed* tweets is 0.14 and 0.11 in each dataset, respectively.

We believe the discrepancies in percentages between both datasets can be attributed to the nature of each dataset, where ANTiVax was collected to study COVID-19's vaccine misinformation specifically, and CMU-MisCov19 was collected to study misinformation related to COVID-19 in general. Hence, ANTiVax may contain more vaccination *experiences* that were shared by users in a narrative style compared to CMU-MisCov19. Also, each dataset was annotated by different annotators and annotation methods.

### 5.2 Engagement with Narrative Tweets in the Presence of Misinformation

In this section, we investigate how narrativity and misinformation affect users' engagement (i.e., likes/favorites and retweets) with tweets. Our alternate hypothesis is that narrativity and misinformation are related to an increase or decrease in the count of likes (favorites) and retweets of a tweet. We control for the number of followers of a tweet's author since this might also affect the engagement with their tweets.

Figure 2 depicts box plots of counts of favorites

| Engagement | Favorite | | | Retweet | | |
|---|---|---|---|---|---|---|
| Dataset | ANTiVax | CMU-MisCov19 | Comb. | ANTiVax | CMU-MisCov19 | Comb. |
| intercept | 3.03 | 3.40 | 3.2 | 1.27 | 2.1 | 1.63 |
| misinfo | -0.47 | 0.64 | -0.12 | 0.35 | 0.36 | 0.21 |
| narrative | 0.10 | 0.28 | 0.52 | -0.43 | -0.24 | -0.34 |
| misinfo x narrative | 0.04 | -3.5 | -0.99 | 0.20 | -2.8 | -0.25 |
| follower count | 3.58 | 5.15 | 5.26 | 4.53 | 5.50 | 5.88 |
| narrative x follower count | 23.92 | -0.93 | 2.26 | 21.73 | -0.82 | 1.46 |
| misinfo x follower count | 4.88 | 17.76 | 11.65 | 3.11 | 17.16 | 9.63 |
| narrative x misinfo x follower count | 1500.48 | 1962.69 | 1546.66 | 1632.43 | 2070.20 | 1679.24 |

Table 5: Coefficients of variables resulting from the Poisson regression GLMs with log link function. All coefficients are significant with $p << 0.01$. Positive (negative) values indicate a positive (negative) association with the outcome variable.

(likes) and retweets after removing outliers. Misinformed tweets receive significantly fewer likes than informed tweets, regardless of their narrativity, across both datasets. Narrative tweets typically received more favorites than their non-narrative counterparts within a given dataset and the absence or presence of misinformation. For the ANTi-Vax dataset, informed narrative tweets have the least number of retweets (zero) compared to the other three categories of tweets in that dataset. For the CMU-MisCov19 dataset, informed non-narrative tweets have a higher number of retweets than the other groups, where both misinformed narrative and non-narrative tweets typically received no retweets.

Next, we built statistical models of the relationships between narrativity, misinformation, follower count, and user engagement. Given that distributions of both likes and retweets follow a Poisson distribution with many tweets having a value of zero likes or retweets, and a heavy right skew for the other values, we utilize generalized linear models (GLM) (Nelder and Wedderburn, 1972) with a Poisson distribution (Adelson, 1966; Thompson, 2001) (as our dependent variables are counts) and the canonical *log* link function (Manning and Mullahy, 2001). We fit six GLM model on the variables: for each dataset (and for both combined), we fit one model with likes as the outcome variable and another with retweets as the outcome. The independent variables are the annotations for misinformation and narrativity, the count of followers, and the interactions between those three main variables.

Table 5 shows the results of the GLMs, with

the set of coefficients for one model listed in each column. We normalized each of the numbers of followers, retweets, and favorites so that they are in the range [0-1]. Similar trends can generally be observed in the cases of favorites and retweets. Tweets that use misinformation had a lower engagement, and tweets with narrative had a higher engagement, but tweets that used narratives in the context of misinformation (narrative x misinfo) had higher engagement only in the ANTiVax dataset. This suggests that false narratives about receiving vaccinations got more engagement than narratives containing misinformation in a general set of COVID-19 tweets, which actually got less engagement. When considering follower count and its interaction with these variables, raw follower count does have a positive relationship with the number of likes or retweets that a tweet gets, and this effect is actually amplified if the tweet contains misinformation (misinfo x follower count) and even more so if the tweet uses narrative style (narrative x misinfo x follower count).

## 5.3 The Language of Narratives and Misinformation

To study the linguistic variations between narrative and non-narrative tweets and between informed and misinformed tweets, we analyzed tweets from each pair of groups using the 2022 version of the Linguistic Inquiry and Word Count (LIWC) program (Boyd et al., 2022). LIWC is a text analysis tool that examines different lexical categories considered psychologically meaningful. It calculates the percentage of words from various lexical categories that are present in a given text. We ran the LIWC analysis on all tweets from the two datasets.

We use the nonparametric independent Mann-Whitney U test (Mann and Whitney, 1947) to establish statistical significance by determining the difference in means. The mean LIWC score for each category served as the test statistic. Since we perform an extensive set of statistical tests, we may find significant differences stemming from type I errors (i.e., false positives). Hence, we apply the Bonferroni correction method (Dunn, 1961) to avoid such errors by setting $\alpha = 0.05/nt$, where $nt$ is the number of tests performed. Then we assume the significance of the difference between means if p-value $< \alpha$ (i.e., accept $H_1$ hypothesis).

We follow three approaches to examine the content of tweets that share narratives: based on as-

| Dataset | ANTiVax | | | CMU-MisCov19 | | | Combined | | |
|---|---|---|---|---|---|---|---|---|---|
| Info or Misinfo | Info | Misinfo | All | Info | Misinfo | All | Info | Misinfo | All |
| LIWC category | R | | | | | | | | |
| WC | 1.22 | 1.12 | - | 1.23 | 1.22 | 1.22 | 1.09 | 1.14 | - |
| Analytic | 0.82 | - | 0.83 | 0.85 | 0.85 | 0.84 | 0.83 | - | 0.83 |
| Authentic | 1.48 | 1.54 | 1.79 | 1.48 | - | 1.51 | 1.6 | 1.54 | 1.83 |
| tone_pos | 2.74 | - | 2.9 | - | - | - | 2.44 | 1.2 | 2.59 |
| tone_neg | 0.52 | 0.8 | 0.55 | - | - | - | 0.54 | 0.79 | 0.57 |
| emo_pos | 5.71 | 2.0 | 7.71 | - | - | 1.71 | 6.22 | 2.06 | 7.63 |
| emo_neg | 0.49 | 0.83 | 0.6 | 0.71 | - | 0.72 | 0.55 | 0.82 | 0.63 |
| health | 1.03 | 0.86 | 1.07 | - | - | - | 1.13 | 0.86 | 1.11 |
| death | - | 1.71 | 0.35 | 1.79 | - | 1.53 | 0.5 | 1.67 | 0.43 |
| function | 1.06 | 1.07 | 1.11 | 1.16 | 1.23 | 1.18 | 1.08 | 1.1 | 1.12 |
| pronoun | 1.38 | 1.25 | 1.5 | 1.62 | 1.53 | 1.61 | 1.44 | 1.29 | 1.52 |
| article | 0.73 | - | 0.73 | - | - | - | 0.79 | - | 0.77 |
| prep | 0.75 | 1.2 | 0.89 | - | - | - | 0.8 | 1.18 | 0.91 |
| auxverb | - | - | 0.85 | - | - | - | 0.87 | - | 0.85 |
| conj | - | 1.4 | 1.14 | - | - | 1.21 | - | 1.4 | 1.13 |
| negate | 0.47 | - | 0.45 | - | - | - | 0.44 | - | 0.46 |
| family | 4.42 | 3.4 | 4.76 | 5.0 | - | 5.13 | 4.67 | 3.43 | 5.06 |
| friend | 1.67 | 3.5 | 2.5 | 4.0 | 4.5 | 4.0 | 1.83 | 4.5 | 2.5 |
| Cognition | 0.59 | 0.88 | 0.6 | - | - | - | 0.6 | 0.89 | 0.63 |
| ppron | 1.55 | 1.73 | 1.97 | 1.95 | 2.04 | 2.0 | 1.72 | 1.77 | 2.03 |
| Time | - | 1.26 | 1.1 | 1.34 | 1.37 | 1.34 | 1.09 | 1.09 | 1.15 |
| cause | - | 0.74 | 0.45 | - | - | - | - | 0.76 | 0.49 |

Table 6: Summary of statistical testing for significance of LIWC categories related to **narratives** using Mann-Whitney U test. $R = \mu_{narr}/\mu_{non}$, the ratio between the means of a LIWC category for narrative and non-narrative tweets, respectively. - denotes non-significant results, i.e., p-value $\geq \alpha$. Blue denotes $R \geq 1$ (i.e., the category is used more in narratives than non-narratives) and Red denotes $R < 1$. WC = word count.

pects of our definition of narrative used in annotation, based on the analysis of Boyd et al. (2020) about how narrative is structurally and linguistically built, and we follow an approach that examines LIWC linguistic dimensions, slightly similar to the approach proposed by Memon and Carley (2020) which examines the linguistic dimensions between Twitter communities that share narratives about COVID-19.

### 5.3.1 Narrative Definition

Dahlstrom (2021) defines a narrative as a triad of character, causality, and temporality, where a character shares a set of related (i.e. causality) events and experiences over a period of time (i.e. temporality). Based on that definition, we utilize LIWC'22 categories that correspond to characters (*pronoun*), causality (*cause*), and temporality (*Time*). In table 6, Narrative tweets show significantly higher usage of pronouns and Time words than non-narrative tweets, which aligns with the definitions. We also note that *family* (e.g., parent, baby, son) and *friend* (e.g., dude, boyfriend) categories, indicative of characters, are used significantly more when in expressing narratives than non-narratives for both datasets. This aligns with

| Dataset | ANTiVax | | | CMU-MisCov19 | | | Combined | | |
|---|---|---|---|---|---|---|---|---|---|
| Narrativity | Non | Narr | All | Non | Narr | All | Non | Narr | All |
| LIWC category | R | | | | | | | | |
| WC | 1.5 | 1.38 | 1.34 | - | - | - | 1.3 | 1.36 | 1.27 |
| Analytic | - | 1.19 | 1.14 | 1.1 | - | 1.11 | - | 1.17 | 1.13 |
| Authentic | 0.63 | 0.66 | 0.52 | 0.81 | - | 0.79 | 0.69 | 0.66 | 0.55 |
| tone_pos | - | 0.35 | 0.39 | - | - | - | - | 0.38 | 0.44 |
| tone_neg | 0.99 | 1.53 | 1.39 | - | - | - | - | 1.46 | 1.3 |
| emo_pos | 0.44 | 0.15 | 0.12 | - | - | - | 0.47 | 0.16 | 0.14 |
| emo_neg | 0.65 | 1.11 | - | - | - | - | 0.74 | 1.1 | - |
| health | 0.89 | 0.74 | 0.86 | - | - | 1.17 | - | 0.75 | 0.9 |
| death | 17.2 | 29.4 | 18.8 | - | - | - | 4.88 | 16.25 | 7.0 |
| Conversation | 1.03 | 1.02 | 1.14 | - | - | - | 1.01 | 1.02 | 1.12 |
| swear | 1.14 | - | 1.07 | 0.38 | - | 0.4 | - | - | - |
| function | 0.9 | 0.91 | 0.88 | - | - | 0.93 | 0.91 | 0.92 | 0.88 |
| pronoun | 0.81 | 0.74 | 0.67 | - | - | - | 0.85 | 0.76 | 0.71 |
| family | - | - | 0.32 | - | - | 0.41 | - | - | 0.33 |
| friend | - | - | 0.22 | - | - | - | - | - | 0.38 |

Table 7: Summary of statistical testing for significance of LIWC categories related to **misinformation** using Mann-Whitney U test. $R = \mu_{misinfo}/\mu_{info}$, the ratio between the means of a LIWC category for misinformed and informed tweets, respectively. - denotes non-significant results, i.e., p-value $\geq \alpha$. Blue denotes $R \geq 1$ (i.e., the category is used more in misinformed tweets than informed ones) and Red denotes $R < 1$.

Pennebaker (2011); Boyd et al. (2020) and the definition of a narrative (Dahlstrom, 2021), where such categories are often used to represent characters. On the other hand, *cause* words show insignificant differences in most tests or a significantly fewer cause words in the ANTiVax dataset. We believe the discrepancy in causality results from the high amount of non-narrative information that explicitly uses causality-related language ("5G causes COVID-19!"), skewing the ratio of casual language away from the narrative class.

### 5.3.2 Narrative Arc

In LIWC'22 (Boyd et al., 2022), Narrative Arc analysis was introduced to understand and measure the text's narrativity (Boyd et al., 2020). The Arc of a narrative is defined as three stages: *1. Staging*, where a storyteller uses more *articles* and *prepositions* to introduce their narrative. *2. Plot Progression*, where staging declines and uses more words – *pronouns, auxiliary verbs, conjunctions, and negations* – which signal events and who is involved in those events and how events progress. *3. Cognitive Tension*, where more *Insight, Causation, Discrepancy, Tentative* and *Certitude* words are used to describe psychological tension and conflict, where characters strife to achieve possible goals. While we found tweets to be too short to use the full narrative arc feature of LIWC'22, we examine the scores for several of the important categories found to be related to narratives.

**Staging**. From table 6, we can see that the use of articles and prepositions significantly differs between narrative and non-narrative tweets for the whole dataset. Where narratives use fewer words of those categories than non-narratives. For CMU-MisCov19, all tests show no significant differences between narrative and non-narratives. For the ANTiVax dataset, only the misinformed tweets show no significant differences in the use of articles and more use of prepositions.

**Plot Progression**. Only pronouns show higher significant usage in narrative tweets. Contrary to pronouns, the other categories show insignificant differences in most tests or significance when fewer words are used in narrative than in non-narrative tweets, except for conjunctions.

**Cognitive Tension**. In LIWC'22, Cognition is a super category that sums the indices of all *insight, causation, discrepancy, tentative, and certitude* subcategories. From table 6, cognition is significantly lower in narrative than non-narrative tweets for the ANTiVax dataset, while it's insignificant for CMU-MisCov19.

We believe that the short length of tweets forced by the platform hinders Twitter users from using more words that would fall under the LIWC categories proposed by Boyd et al. (2020) which were intended to measure narrative arcs in longer stories. This also justifies the discrepancies between our results and those of Boyd et al. (2020), where they concluded those categories after analyzing a large set of corpora collected from long novels, film transcripts, short stories, etc., where the average word count ranges from a few hundred to tens of thousands of words.

### 5.3.3 Linguistic Dimensions of Narratives and Misinformation

**Analytical and Authentic**. The LIWC categories *Analytic* and *Authentic* are two summary variables that measure 1) logical and formal thinking and 2) perceived honesty and genuineness, respectively. We observe from table 6, that narratives have significantly lower analytic and higher authentic scores than non-narratives in both datasets. This also aligns with Boyd et al. (2022); Memon and Carley (2020). Additionally, we notice from table 7 that misinformed tweets are less honest and more analytical in both datasets than informed tweets.

**Tones and Emotions**. Tone measures how positive or negative a sentiment a piece of text has. LIWC, calculate two categories *tone_pos* and

*tone_neg*, where the higher the score, the more positive or negative the tone is, respectively. Emotion measures how much a text has positive or negative emotions (e.g., anxiety, anger, sadness) that correspond to the LIWC categories *emo_pos* and *emo_neg*, respectively. According to table 6, narrative tweets significantly share more positive tones and emotions than non-narrative tweets in ANTiVax, while it's insignificant in almost all cases for CMU-MisCov19. While negative tones and emotions are more commonly shared among non-narrative tweets. Table 7 shows that in ANTiVax dataset, misinformation was not as clearly related to tone, though in cases where significant results were found, positive tone and emotion were used less in tweets that contained misinformation. While for CMU-MisCov19, it is all insignificant.

**Health and Death**. The *health* LIWC category contains 715 words related to health, illness, wellness, and mental health (e.g., medic, patient, hospital, gym, ... etc). The *death* category has 109 words (e.g., dead, kill, ... etc). From table 6, we find that health words are overall significantly used more in narrative than non-narrative tweets, while death words are overall used more in non-narrative tweets, and that might be attributed to having many tweets where news agencies sharing COVID-19-related deaths in a non-narrative formal style. Investigating how health and death are shared among informed and misinformed tweets, we find in table 7 that the CMU-MisCov19 dataset mainly shows insignificant differences in means. While for the ANTiVax dataset, misinformed tweets share fewer health words and many more death words than informed tweets, which shows that COVID-19 vaccine-related misinformation is often focused on death and negative side effects.

## 6 Thematic Analysis

In order to understand the prevalent themes within the discourse surrounding COVID-19 vaccinations on social media, we conducted a manual thematic analysis on a selected sample of 400 tweets, equally balanced across each of the following categories: narrative information, non-narrative information, narrative misinformation, and non-narrative misinformation (100 tweets from each category). We first examined the sampled tweets to uncover the recurring themes, and then labeled each tweet according to the predominant theme present. Table 8 presents the most common themes within each sub-

| | Narrative | | Non-Narrative | |
|---|---|---|---|---|
| **Informative** | 1. got vaccinated and feeling positive
2. feeling/anticipating vaccine side effects
3. friends/family member received the shot/feeling positive
4. eager to receive vaccine
5. someone got covid
6. vaccine scheduling
7. friends/family member vaccine scheduled/feeling positive
8. expressing gratitude towards healthcare workers
9. other | 39%
15%
11%
10%
3%
3%
3%
3%
13% | 1. don't worry about what's in the vaccine
2. get vaccinated
3. sharing news
4. questioning/debunking conspiracy theories
5. frustration towards those not following guidelines
6. sharing local information
7. vaccine efficacy
8. mislabeled - should be narrative
9. public health tips
10. joke
11. criticizing government
12. other | 30%
13%
9%
7%
6%
6%
6%
4%
3%
3%
3%
10% |
| **Misinformative** | 1. conspiracy theory
2. vaccine related deaths and side effects
3. vaccine hesitancy
4. covid home remedy
5. other | 38%
28%
20%
6%
8% | 1. suspecting vaccine as gene therapy
2. vaccine hesitancy and concerns
3. covid as a bioweapon
4. govt/large corp
5. covid is a depopulation agenda
6. covid home remedy
7. other | 19%
19%
18%
14%
13%
10%
7% |

Table 8: Results from the thematic analysis of COVID-19 Vaccine Narratives and Non-narratives and their proportions of the total dataset.

set, along with their respective percentages within the subset. Appendix E presents detailed thematic analysis results with paraphrased examples.

We find that a majority of the **informative narratives** described people's experiences with getting the vaccine, or expressing excitement that a loved one was vaccinated. This focus on vaccination-related stories can be partially attributed to the larger size of the *ANTiVax*, which is specifically focused on vaccine-related information and misinformation, relative to the *CMU-MisCov19* dataset, which is more generally about COVID-19. However, even across both datasets, we find that many users' stories centered around either vaccine experiences or experiences with COVID-19 itself, as these were some the main ways that everyday users were personally affected by the pandemic. **Misinformative narratives** typically centered around storylines that provided evidence for conspiracy theories or harmful side effects of the vaccine, and were less often written as first-person narratives. When users did talk about themselves in the misinformative narrative category, the stories described trying home remedies or hypothetical stories justifying why they would not get vaccinated.

**Non-narrative informative** (i.e., those not labeled as misinformation or narrative) tweets contained a wide range of topics, from sharing news and public health tips to memes in the form "If X, don't worry about what's in the vaccine", where X is something commonly done yet potentially unhealthy, such as eating fast food. These tweets are meant to downplay concerns others may be expressing about getting vaccinated. **Misinformative**

**non-narratives** presented similar ideas to the misinformative narratives, but these were presented in the form of "facts" about the dangers of vaccines or supporting conspiracy theories, rather than stories.

## 7 Conclusion

In this paper, we study the relationship between narrative communication and health misinformation on social media, specifically in the context of Twitter. We manually labeled a subset of tweets from misinformation datasets to add labels for narrative, and trained models to extend these labels to the entire datasets. Then, using statistical modeling, we find that narrativity is connected to higher user engagement and misinformation to less engagement, but narrativity may help increase engagement of misinformation in some contexts such as for users with many followers. This finding may have implications for better promoting accurate health information by presenting and amplifying narrative messages. In line with studies of other domains, narratives showed more authenticity, had more mentions of people, time and many more expressions of positive emotion, while misinformation was less authentic used more analytic language, and had dramatically more references to death. Many narratives focused on experiences of vaccines, illness, or home remedies, while non-narratives focused more on news updates, public health information, and conspiracy theories. These findings shed light on how both the style and content of messages can relate to how users engage with then, suggesting that both are important when seeking to combat misinformation.

## 8 Limitations

As we used English-language data from Twitter during the COVID-19 pandemic, the language present in the dataset will be biased to this particular population and may limit the ability to extrapolate conclusions about narrativity and misinformation to other contexts. For example, given the universality of the pandemic, everyday users may have already been more invested in searching for and understanding health related information than they would have for other health issues. In the future, research should focus on other languages, platforms, and health contexts.

Further, the tweets were rehydrated based on their IDs, and some portion of the tweets were no longer available, disproportionately affecting the misinformation-related tweets (and potentially some of the higher-engagement misinformation tweets, since those may have been more likely to be debunked and subsequently deleted). In a less transparent way, Twitter's own content moderation policy also likely affected how much misinformation was shown to users, decreasing engagement, though we are unsure to what extent. Lastly, with recent changes to the Twitter (recently re-branded as "X") it is unclear how available Twitter-based datasets will be in the future [3].

This paper uses the LIWC (Language Inquiry with Word Count) tool, which, though very popular, only uses surface-level analysis in the form of word count analysis and fails to provide an in-depth interpretation of the data. Despite participating in three rounds of training sessions, the annotators still encountered several disagreements among each other which were resolved through discussion. With additional training, it may be possible to achieve an even higher inter-coder agreement and more reliable annotations.

Additionally, the narrative labels used for the full dataset are based on the predictions of a deep-learning model with imperfect predictive power, and if errors are made in a systematic with respect to the relationship between narrative and misinformation, some of the later analyses could be slightly skewed by this. We illustrate the extent of some possible differences in Appendix D, showing that the overall conclusions are unlikely to change.

---

[3]Those seeking to access the dataset for replication purposes are encouraged to reach out to the authors of the paper for a discussion of how data may or may not be shared based on the evolution of the current situation and best practices.

## Ethics Statement

In this study, only pre-released Twitter datasets were used and collected via the official Twitter API using their IDs. Tweets that had been removed by their authors or by Twitter were not used in the study. All example tweets presented in the paper are paraphrased in order to preserve the anonymity of their original authors and in accordance, and data will only be shared as a list of tweet IDs with our own additional annotations. The annotators used in the study were all authors and all are employed in research-related positions which compensate them for their work. While it is possible that the findings in this paper could be leveraged in order to help spread misinformation through the effective use of narratives, we do not provide any causal connections between narrativity and engagement nor do we aim to provide specific recommendations about how to increase tweet engagement, and this work may also help to provide new lenses through which to study misinformation and its spread in order to better combat it.

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

## A  Annotation Guidelines

Below are the guidelines developed during the annotation process and agreed upon by the annotators. Note that these guidelines were specifically developed for annotating *tweets* and should not be directly used annotating other types of text as narrative before considering and adapting to other contexts and media. All examples provided are related to vaccines and/or COVID-19.

### A.1  Narrative definitions

According to (Kreuter et al., 2007), a narrative is defined as a representation of connected events and characters that has an identifiable structure, is bounded in space and time, and contains implicit or explicit messages about the topic being addressed.

(Bilandzic and Busselle, 2012b) say that a narrative refers to a presentation of an event(s) experienced by a specific character(s) in a setting.

And according to (Dahlstrom, 2021), a narrative is defined as a message that describes the experience of specific characters across a series of related events over a defined time period—a triumvirate of character, causality, and temporality. At its core, narrative is the telling of someone's experience about something.

## A.2 Rules

Based on the definitions provided, the rules for labeling the tweets are as follows: the tweet must contain (1) at least one specific character (normally is a person) who experiences (2) a series of related events. You may assume that the presence of multiple events implies temporality and do not need to specifically check for temporality during annotation.

### A.2.1 Characters

Character/characters need to refer to specific individuals. Characters can be the author of the text (1st person), but can also be someone else who is mentioned in the text (2nd or 3rd person).

*Examples*: "Scientist", "CDC", any mentions like "this journalist", "14 people" (where this refers to a specific set of individuals that you could hypothetically identify) etc., can be considered characters. "People", "humanity", "the world", etc. don't count as characters. 'You' or 'your' (and they) can also be used in a generic way that does not refer to a specific person and should therefore only be considered a character if it directly refers to someone specific such as "Joe Biden, you should consider giving more free covid tests". "We" and "us" can be a character because it includes the author. Mentions of a country or country's government in general ("The U.S.") are usually too vague to be considered a character, but identifiable individuals ("The United States Supreme Court") within the government could be characters.

### A.2.2 Events

Emotions, thoughts, or other non-observable actions can be considered an event. Emojis and punctuation can indicate emotions. The characters involved don't necessarily need to take any actions, but should be involved in or experiencing the events somehow. Events can be fictional, false, or occurring in the future. They don't need to be actual things that have definitely happened.

*Examples*: "I am so happy that I got the covid vaccine" can be a narrative because the person is happy now and the vaccine is in the past, so there is temporality and causality. Only saying "I got the vaccine" is not a narrative for our study because it does not reflect a *series of related events*.

### A.2.3 Additional cases to consider

Hypothetical situations using an if-then format, such as "if I had been to Dr. Dennis as a child, then I would be naturally immune to covid" are not narratives. It could be rewritten as "I went to Dr. Dennis as a child, and now I have natural immunity to covid" which conveys the same information, but in more of a narrative form. The narrative is not only about what information is being presented, but the *style* of communication being used (*how* it is presented). Quotes like "CDC director said 'there are 1000 new covid cases'" is not a narrative if it is just mentioning that someone said a fact but there is no story. But if the CDC director says "I have been sick for three days and the medicine didn't have any effect on me... " this can be considered a narrative. Similarly, narratives within jokes/non-serious texts are still considered narratives. E.g., "Bill Gates is going to come to my house to get the tracker out of my arm from the covid vaccine." This is a narrative. Advice is not a narrative: "You should get your covid vaccine". News headlines can be narratives, but they are not always narratives. For example, "Vaccine Mass Sterilization Depopulation Agenda Revealed on Amazon 'Utopia' Show" is not a narrative. In general, try not to overthink it and perform mental gymnastics to determine how a tweet might possibly meet the criteria. Ask yourself: does this tweet tell a story about someone's experience of something? If so, it is a narrative.

## A.3 Examples of difficult cases

During pilot rounds of annotation, the following cases were difficult to come to an agreement about. After further discussion with communication research experts, the following decisions and rationales were made and provided as guidance for the final annotation process. Note that these were some of the most difficult-to-annotate (according to the annotators) edge cases, and are not representative of a majority of the dataset.

1. New Jersey has launched a website to debunk rumors and hoaxes associated with the spread of the coronavirus, following a false text message of impending national lockdown that circulated widely across the U.S.

Label: Not a narrative

Reason: New Jersey [government] is not a character because it is too vague.

2. CIA knew about #coronavirus #BioWeapons before Chinese health authorities #USNews #COVID19 was released by #DonaldTrump

Label: Narrative

Reason: There are two events that are connected: CIA knowing about bioweapons before Chinese health authorities, and Donald Trump releasing COVID-19.

3. Me thinks so too! The gov can't say they have been poisoning us for years - imagine the uproar - it affects us all - eating synthetic foods for decades. Why does supermarket meat taste like Rubber? coronvirus smokescreen.. ..but then there is COVID-19 is that the bioweapon?

Label: Narrative

Reason: This tells someone's (the tweet author's) experience of something: the government poisoning food and supermarket meat tasting like rubber.

4. Got my first vaccine dose today!

Label: Narrative

Reason: The ! at the end conveys an emotion that the author is experiencing after getting the vaccine.

5. #vaccinated can I fly on #americanairlines after the second COVID-19 vaccine?

Label: Narrative

Reason: The author expresses that they got their second COVID-19 vaccine (#vaccinated) and is now wondering about the possibility of air travel.

6. I have had my money on Oxford-AstraZeneca from day 1. So excited to hear this news!

Label: Narrative

Reason: The author has been hoping Oxford-AstraZeneca would be successful and is now expressing excitement about the fact that they were.

7. @RudyGiuliani Many of us have taken this drug for Malaria prevention but never while suffering from COVID19. You are misleading the public with dangerous misinformation

Label: Narrative

Reason: This tweet author "speaks" directly to Rudy Giuliani. The author is one of the "many of us" who has taken the drug.

## B Hyperparameter Tuning and Preprocessing

For SVM classification, we used a grid search over both kernel type (linear, polynomial, or RBF) and degree (1, 2, and 3), with a polynomial kernel and a degree of 1 giving the best results on the validation set. For logistic regression, we considered $C = \{1, 2, 3\}$ and $C = 2$ led to the best results.

For the bag-of-words models, we also experimented with various pre-processing techniques, including lowercasing, lemmatization, removal of URLs, stop words, and hashtags. We used a combination of these techniques along with hyperparameter tuning that gave us the best performance on the validation set and used that model to predict the labels for the test set. The best results for all bag-of-words models were achieved when removing stopwords, URLs, and hashtag symbols ('#') while keeping the text of the hashtag itself.

For the transformer-based deep learning models, we initialize each from the model checkpoint and fine-tune on our training data for 5 epochs with a batch size of 16, weight decay of 0.01, and a learning rate of 2e-5.

## C Few-Shot Experiments

This section presents detailed results on the effect of the number of in-context examples that are provided to the GPT models for few-shot learning. In each case, we balance the number of narrative and non-narrative examples from each of the two source datasets, meaning that our number of training examples, $k$ is always multiple of 4 to maintain this balance. From results on our development set, we find that the overall best results occur when $k = 20$, and therefore we select this value for the rest of our experiments in the paper. This means that we have $n = 5$ narratives from each dataset and $n = 5$ non-narratives from each of the two datasets. The results for the GPT-3 Davinci model are presented in Table 9, and the results for the GPT-3.5-Turbo model are presented in Table 10.

| GPT-3 (Davinci) | | | | | | |
|---|---|---|---|---|---|---|
| **Model** | **k=4** | **k=8** | **k=12** | **k=16** | **k=20** | **k=40** |
| few-shot | 0.579 | 0.603 | 0.612 | 0.646 | **0.699** | 0.677 |
| +def | 0.581 | 0.594 | 0.638 | 0.643 | 0.697 | **0.700** |
| +inst | 0.560 | 0.629 | 0.630 | 0.666 | **0.698** | 0.642 |
| +def+inst | 0.601 | 0.638 | 0.680 | 0.689 | **0.712** | 0.685 |

Table 9: Narrative class F1 scores *averaged across three runs* for GPT-3 models across various few-shot settings, trained on detecting narratives using the validation set only. **Bold** indicates the top F1 score per model.

| GPT-3.5-Turbo | | | | | | |
|---|---|---|---|---|---|---|
| **Model** | **k=4** | **k=8** | **k=12** | **k=16** | **k=20** | **k=40** |
| few-shot | 0.539 | 0.548 | 0.542 | 0.565 | **0.584** | 0.568 |
| +def | 0.566 | 0.580 | 0.643 | 0.631 | **0.675** | 0.564 |
| +inst | 0.571 | 0.591 | 0.597 | 0.616 | **0.642** | 0.575 |
| +def+inst | 0.596 | 0.611 | **0.648** | 0.644 | 0.636 | 0.590 |

Table 10: Narrative class F1 scores *averaged across three runs* for GPT-3.5-Turbo models across various few-shot settings, trained on detecting narratives using the validation set only. **Bold** indicates the top F1 score per model.

# D  LIWC analysis of manual-annotations versus machine-annotations

In this appendix, we computed the LIWC scores similar to both tables 6 and 7 for only the 3,000 tweets that were manually annotated. See tables 11 and 12. We found that there is no meaningful difference between the LIWC scores of the 3000 tweets annotated by the authors and all the 14561 tweets annotated by the best-performing model (RoBERTa). The means of differences of LIWC scores between human-annotated data and model-annotated data are 0.225 and 0.282 with standard deviation of 0.601 and 0.927 for all values in tables 6 and 11, and tables 7 and 12, respectively. These small differences reflect the strength of the best-performing RoBERTa model that achieved an F1-score of 0.924, and we did not identify any values that contradict the claims made in the paper. However, we did notice that fewer claims could be reliably made given the smaller size of the data when using only 3,000 versus 14,561 texts. The number of LIWC scores that were not statistically significant in human-annotated data only, but were statistically significant in the full human+model-annotated data, is 40 scores and 28 scores for tables 6 and 7 respectively. This can be attributed to the increased size of the dataset, making a larger number of the results statistically significant while still controlling for multiple comparisons.

| Dataset | ANTiVax | | | CMU-MisCov19 | | | Combined | | |
|---|---|---|---|---|---|---|---|---|---|
| Info or Misinfo | Info | Misinfo | All | Info | Misinfo | All | Info | Misinfo | All |
| LIWC category | R | | | | | | | | |
| WC | 1.22 | - | - | 1.25 | - | 1.26 | 1.12 | 1.15 | - |
| Analytic | 0.83 | - | 0.84 | - | - | - | 0.84 | - | 0.84 |
| Authentic | 1.58 | - | 1.85 | - | - | - | 1.68 | - | 1.85 |
| tone_pos | 2.29 | - | 2.5 | - | - | - | 2.17 | - | 2.29 |
| tone_neg | 0.54 | - | 0.57 | - | - | - | 0.54 | - | 0.57 |
| emo_pos | 3.62 | - | 5.17 | - | - | - | 3.87 | - | 5.16 |
| emo_neg | 0.49 | - | 0.6 | 0.54 | - | 0.56 | 0.53 | - | 0.61 |
| health | - | - | - | - | - | - | - | - | - |
| death | - | - | 0.44 | - | - | - | - | - | 0.5 |
| function | - | - | 1.09 | 1.24 | - | 1.26 | 1.08 | - | 1.11 |
| pronoun | 1.27 | - | 1.41 | 1.69 | - | 1.68 | 1.34 | - | 1.44 |
| article | 0.78 | - | 0.75 | - | - | - | 0.84 | - | 0.8 |
| prep | 0.78 | - | 0.89 | - | - | - | 0.82 | 1.2 | 0.91 |
| auxverb | - | - | 0.84 | - | - | - | - | - | 0.85 |
| conj | - | 1.35 | - | - | - | - | - | 1.38 | - |
| negate | 0.48 | - | 0.47 | - | - | - | 0.45 | - | 0.49 |
| family | 3.04 | 4.67 | 3.71 | - | - | 4.16 | 3.2 | 4.57 | 3.9 |
| friend | - | - | - | - | 5.0 | - | - | - | 4.0 |
| Cognition | 0.65 | - | 0.64 | - | - | - | 0.64 | - | 0.65 |
| ppron | 1.59 | 1.74 | 1.97 | 2.04 | - | 2.07 | 1.72 | 1.76 | 2.02 |
| Time | - | - | - | 1.54 | - | 1.46 | - | - | 1.12 |
| cause | - | - | 0.49 | - | - | - | - | - | 0.52 |

Table 11: Summary of statistical testing for significance of LIWC categories of the 3,000 manually annotated tweets related to **narratives** using Mann-Whitney U test. $R = \mu_{narr}/\mu_{non}$, the ratio between the means of a LIWC category for narrative and non-narrative tweets, respectively. - denotes non-significant results, i.e., p-value $\geq \alpha$. Blue denotes $R \geq 1$ (i.e., the category is used more in narratives than non-narratives) and Red denotes $R < 1$. WC = word count.

| Dataset | ANTiVax | | | CMU-MisCov19 | | | Combined | | |
|---|---|---|---|---|---|---|---|---|---|
| Narrativity | Non | Narr | All | Non | Narr | All | Non | Narr | All |
| LIWC category | R | | | | | | | | |
| WC | 1.46 | 1.35 | 1.31 | - | - | - | 1.29 | 1.33 | 1.24 |
| Analytic | - | 1.22 | 1.14 | - | - | 1.16 | - | 1.24 | 1.14 |
| Authentic | 0.64 | 0.54 | 0.49 | - | - | - | 0.71 | 0.55 | 0.54 |
| tone_pos | - | 0.31 | 0.39 | - | - | - | - | 0.32 | 0.45 |
| tone_neg | - | 1.52 | 1.37 | - | - | - | - | 1.44 | 1.29 |
| emo_pos | - | 0.09 | 0.12 | - | - | - | 0.34 | 0.09 | 0.13 |
| emo_neg | 0.67 | - | - | - | - | - | 0.74 | - | - |
| health | - | - | 0.86 | - | - | - | - | - | - |
| death | 21.0 | 24.17 | 18.6 | - | - | - | 6.0 | 14.44 | 7.73 |
| Conversation | - | - | 1.19 | - | - | - | - | - | 1.15 |
| swear | - | - | - | - | - | - | - | - | - |
| function | - | - | 0.91 | - | - | - | - | - | 0.9 |
| pronoun | 0.77 | 0.74 | 0.68 | - | - | - | 0.79 | 0.74 | 0.69 |
| family | - | - | 0.39 | - | - | - | - | - | 0.38 |
| friend | - | - | - | - | - | - | - | - | - |

Table 12: Summary of statistical testing for significance of LIWC categories of the 3,000 manually annotated tweets related to **misinformation** using Mann-Whitney U test. $R = \mu_{misinfo}/\mu_{info}$, the ratio between the means of a LIWC category for misinformed and informed tweets, respectively. - denotes non-significant results, i.e., p-value $\geq \alpha$. Blue denotes $R \geq 1$ (i.e., the category is used more in misinformed tweets than informed ones) and Red denotes $R < 1$.

# E  Thematic Analysis with Examples

In this section of the appendix, we present a breakdown of our thematic analysis results of COVID-19 tweets. Here each subset of the analysis is pre-

| Theme | % | Example |
|---|---|---|
| suspecting vaccine as gene therapy | 19 | Quick reminder, this isn't any normal vaccine. THIS IS GENE THERAPY, which will make a human forever slave to the big tech |
| vaccine hesitancy and concerns | 19 | They be calling them idiots for not wanting a rushed untested vaccine, when it's the vaccine that causes infertility |
| covid is a bioweapon | 18 | Wake up!! This is not a virus, it's a carefully launched bioweapon. |
| govt/large corp | 14 | Fauci knows about the man-made virus long ago, it's all a cover up by the govt to keep the public under their shoes. |
| covid is a depopulation agenda | 13 | There are 10 million unplanned pregnancy terminations since the beginning of this virus. Don't be stupid, this is a depopulation attempt by Bill Gates and the govts to control population growth. |
| covid home remedy | 10 | Take mint leaves, honey, garlic and onion to prevent covid. you don't need the vaccine!!!!!!!!! |
| other | 7 | Sign this petition people! covid is hoax. #stopdepopulation #novax |

Table 13: Thematic Analysis of **Non-Narrative Misinformation** on COVID-19: Predominant Themes, Proportions, and Representative paraphrased Examples

| Theme | % | Example |
|---|---|---|
| don't worry about what's in the vaccine | 30 | If you've ever been to that KFC on the street, don't worry about what's in the vaccine |
| get vaccinated | 13 | Will be getting vaccine next week. _/_ |
| sharing news | 9 | The government will now begin vaccinating kids below the age of 10, starting Wednesday |
| questioning/debunking conspiracy theories | 7 | 5G towers don't cause covid and there is no scientific evidence to support that claim. Stop spreading misinformation and please follow cdc guidelines and save lives |
| frustration towards those not following guidelines | 6 | So frustrating to see selfish people not follow rules. The pandemic isn't over yet. Please stay home and follow the rules. |
| sharing local information | 6 | vaccine information: vaccinations available for ages 14+ this Sunday 3/09 at ABC Pharmacy in Carlton. If interested please call their number at 999-999-9999 #stay vaccinated |
| vaccine efficacy | 6 | The vaccine protects you and your loved ones from this deadly virus. Please believe in science and get vaccinated and save lives. |
| mislabeled - should be narrative | 4 | Hurrayyyy! I am getting vaccinated in two days! Let's go guys! |
| public health tips | 3 | Stay indoors, wear masks and always sanitize your hands when you touch public areas. |
| joke | 3 | my American friend says covid is a planned agenda, but what about Bill Gates acquiring large areas of farm lands?LOL |
| criticizing government | 3 | Only 6% of the adults have received both doses. This is bad governance at its best. You should learn from South Korea |
| other | 10 | My sister argues with me about this vaccine everyday! |

Table 14: Thematic Analysis of **Non-Narrative Information** on COVID-19: Predominant Themes, Proportions, and Representative paraphrased Examples

sented individually with paraphrased examples, included to preserve the anonymity of the authors while still capturing the main types of messages that existed in each theme. Discussion of the themes was originally presented in Section 6.

Table 13 shows the predominant themes occurring in the non-narrative misinformation subset. These are the tweets that were originally labeled as misinformation from the source datasets, and labeled as narratives through our methodology. The themes presented in this table predominantly re-

volve around various conspiracy theories and misinformation regarding COVID-19 and the vaccines. They touch upon unfounded claims such as the virus being a bioweapon, and concerns about the vaccine's safety and potential side effects.

Table 14 describes themes from the non-narrative information category, and predominantly captures the various facets of public sentiment and discourse regarding COVID-19 but in a non-narrative style. The themes range from humorous takes and encouragement for vaccination to the

| Theme | % | Example |
|---|---|---|
| conspiracy theory | 38 | Fauci knew that the virus has been cooking in the lab since 2005 but he covered it up and now Bill Gates is inserting chips in our body through vaccine to track us |
| vaccine related deaths and side effects | 28 | "A boy in Netherlands found dead 30 hours after taking his second covid shot. He was fit until a day before he took that damn vaccine!!!! |
| vaccine hesitancy | 20 | I will not take that vaccine until i get pregnant and give birth to a baby. I don't want to lose my fertility to a manufactured virus and supposed vaccine -_- |
| covid home remedy | 6 | I have been eating garlic, one table spoon of honey, some onions before i go to bed and let me tell you, i have not had covid since! This totally works! and is the best solution for the virus. |
| other | 8 | I unfriended a good friend of mine yesterday because she's totally into this 5G conspiracy theory shit. I don't have any regrets |

Table 15: Thematic Analysis of **Narrative Misinformation** on COVID-19: Predominant Themes, Proportions, and Representative paraphrased Examples

| Theme | % | Example |
|---|---|---|
| got vaccinated and feeling positive | 39 | I got my second shot yesterday. I can finally go visit my mom and dad this weekend and i feel so grateful! <3 |
| feeling/anticipating vaccine side effects | 15 | Got the jab two days ago and i am really feeling the effects. My arm is still sore and i feel the chills too. Not going to work tomorrow |
| friends/family member received the shot/ feeling positive | 11 | My mom and dad got their first shot today and i feel so relived. Let's defeat this pandemic together. Peace |
| eager to receive vaccine | 10 | I could finally book my appointment for the vaccine. I am excited and can't wait to get the shot and meet my friends |
| someone got covid | 3 | Robert did not want to get vaccinated and guess what, he caught the virus and he's sick! |
| vaccine scheduling | 3 | I have been trying to book vaccine appointment for weeks and still haven't made it, i feel so disappointed. Urghhhh!!!!! |
| friends/family member vaccine scheduled/ feeling positive | 3 | My 78yo grandmother could finally book vaccine appointment for tomorrow. I feel so happy and grateful!! |
| expressing gratitude towards healthcare workers | 3 | i got my first shot yesterday and I thank all the frontline workers who have done phenomenally well during this pandemic. You saved countless lives. Feeling grateful!! |
| other | 13 | (includes themes like "criticizing govt's handling of covid", "a joke", "setting example for others by getting vaccinated" etc.) |

Table 16: Thematic Analysis of **Narrative Information** on COVID-19: Predominant Themes, Proportions, and Representative paraphrased Examples

active debunking of conspiracy theories, and so on.

Table 15 highlight various forms of misinformation and misconceptions surrounding COVID-19 and its vaccines which were presented using narrative style. They capture sentiments from deeply entrenched conspiracy theories and fears related to vaccine side effects. The nature of these themes present the challenge of battling misinformation in the era of the pandemic.

Finally, in the themes presented in Table 16 revolve around personal experiences, emotions, and perspectives related to the COVID-19 vaccination process, which were presented in narrative style. These themes capture sentiments ranging from gratitude and optimism after receiving the vaccine to challenges faced during the vaccine scheduling and concerns about side effects.