# OpenReview forum: "Narrative Style and the Spread of Health Misinformation on Twitter"
_EMNLP/2023/Conference — EMNLP 2023 Findings_

### Official Review · Reviewer_93eC · 2023-08-01

**Soundness:** 3

**Excitement:**

3: Ambivalent: It has merits (e.g., it reports state-of-the-art results, the idea is nice), but there are key weaknesses (e.g., it describes incremental work), and it can significantly benefit from another round of revision. However, I won't object to accepting it if my co-reviewers champion it.

**Paper Topic And Main Contributions:**

This paper proposes a system for automatic narrative detection with an application of characterizing the use of narrative in health misinformation on Twitter. The authors manually annotated an existing misinformation tweet dataset for the presence of narrative. They then built text classifiers to predict narrative style, and found that a fine-tuned RoBERTa model performed the best. Using that classifier to label the rest of the dataset, they explore different properties of the relationship between narrative style and misinformation. They find that narrative style is related to higher engagement and that for one of their datasets, narrative is associated with higher engagement with misinformation.

**Reasons To Accept:**

The paper provides a good motivation that narrative is an issue that needs to be addressed with misinformation and takes first steps toward a computational and quantitative analysis.

**Reasons To Reject:**

Since the RoBERTa classifier chosen to annotate the reset of the dataset surely has some systematic errors, it would be more convincing to present results with respect to engagement (section 5) also just for the subset of manually annotated tweets to verify that there is no significant difference between those and the tweets that were annotated for narrative with the classifier.

Otherwise, this paper would benefit most from a more thorough discussion and interpretation of all the results, which as it currently stand are difficult to tie together and so lack coherent findings that could benefit future work.

The results from the GLM seem inconclusive on the relationship between misinformation and narrative. As noted, there is a positive relationship for misinfo x narrative for ANTiVax but not CMU-MisCov19. The paper says that this indicates that false narratives about getting the vaccine get more engagement than other misinformation narratives (lines 433-435). Examples of those other types of misinformation narratives would be informative. I assume false narratives about getting the vaccine would be narratives about people getting the vaccine and then terrible things happening to them? Examples here would be valuable as well.

Effect sizes or something else may help readers interpret the strength of these relationships, which is currently lacking. There is a positive relationship for both datasets for narrative x misinfo x follower count, but with very small coefficient values. The range of coefficient values varies dramatically overall (the -3.5 coefficient for misinfo x narrative for CMU-MisCov19 has a particularly large absolute value, which may be worth talking about).

A more in-depth discussion of the results is needed to properly interpret and contextualize all of the results given. For example, it seems counterintuitive and interesting that there is more narrativity with informed tweets (if I'm reading section 5.1 right). What might be evidence from the tweets to explain this? Are people just sharing stories of getting the vaccine? In general, more interpretation of main themes coming from results in each subsection of section 5 would be helpful.

The LIWC analysis in section 5.3.3 seems to yield unsurprising results when it comes to what terms narratives would use more. More informative would be differences between narrative info and narrative misinfo, but the only difference detected is a focus on death terms. Unsurprising results are okay in general, and maybe the focus on death can contribute to what is not known about COVID misinformation (it is perhaps interesting that the informed tweets are not talking about avoiding death through taking the vaccine, for example), but more interpretation and discussion would be needed. The narrative arc analysis in section 5.3.2 doesn't seem to add any value, since as noted, tweets just seem too short for this tool to work properly and it seems like most of the results are statistically insignificant.

**Reproducibility:**

4: Could mostly reproduce the results, but there may be some variation because of sample variance or minor variations in their interpretation of the protocol or method.

**Reviewer Confidence:**

4: Quite sure. I tried to check the important points carefully. It's unlikely, though conceivable, that I missed something that should affect my ratings.

**Typos Grammar Style And Presentation Improvements:**

* What does the Hydrated % row refer to in Table 1?
* Figure 1 text is too small to easily read, other figures like Fig 2 and tables too
* Line 338 , instead of .
* Line 417: model -> models
* Line 632: authentic used -> authentic, used

---

> ### Author Rebuttal · Authors · 2023-08-29
>
> Thank you for taking the time to read and engage with our work! We appreciated your insights and suggestions. Below we have done our best to address all of your questions and concerns.
>
> 1) Regarding the concern about systematic errors: as you have suggested, we recomputed the LIWC scores in both tables 6 and 7 for only the 3000 tweets that were manually annotated. We will report these results in the appendix so that readers may compare the full set of values directly. We found that there is no meaningful difference between the LIWC scores of the 3000 tweets annotated by the authors and all the 14561 tweets annotated by the best-performing model (RoBERTa). The means of differences of LIWC scores between human-annotated data and model-annotated data are 0.225 and 0.282 with standard deviation of 0.601 and  0.927 for all values in tables 6 and 7, respectively. These small differences reflect the strength of the best-performing RoBERTa model that achieved an F1-score of 0.924, and we did not identify any values that contradict the claims made in the paper. However, we did notice that fewer claims could be reliably made given the smaller size of the data when using only 3000 vs 14561 texts. The number of LIWC scores that were not statistically significant in human-annotated data only, but were statistically significant in the full human+model-annotated data, is 40 scores and 28 scores for tables 6 and 7 respectively. This can be attributed to the increased size of the dataset, making a larger number of the results statistically significant while still controlling for multiple comparisons.
>
> 2) Regarding the small coefficients in the GLMs, many thanks for pointing this out. After normalizing the follower count so that it is in the range 0-1, better matching the other variables which are boolean, we found it retained the same trend reported in the paper for the follower count and interaction terms that use the follower count. However, we agree with your suggestion and find these normalized numbers are more easily comparable and interpretable, so we will update with those numbers in the final version of the paper if accepted.
>
> 3) We will also be eager to expand on the discussion section given an extra page if the paper is accepted. We believe there are useful insights regarding the potential for narratives to amplify the spread of information on Twitter and will add a specific subsection dedicated to "recommendations" for future work in this area.
>
> 4) Regarding your request for examples: we chose not show the exact examples of tweets based on prior scholarly works such as the work by Fiesle and Proferes: “Participant” perceptions of Twitter research ethics." Social Media+ Society 4.1 (2018): 2056305118763366. which suggests that users prefer not to have their exact messages presented in research papers or being used as examples. Our commitment to not directly showcasing examples stems from our ethical stance to safeguard the privacy of the authors of the tweets. However, based on your helpful suggestion, we conducted a thematic analysis of the data in each category (narrative info, narrative misinfo, non-narrative info, non-narrative misinfo), which summarizes the main themes that were present in the tweets so that you can understand the general content of them, along with some paraphrased examples. These results will be added to the paper if accepted and we have included them at the end of this rebuttal (see item 6) for your reference.
>
> 5) Regarding "The narrative arc analysis in section 5.3.2 doesn't seem to add any value" we mainly included this in order to compare to the tool proposed by Boyd et al. which was used to analyze narratives in prior work, since it is one of the only existing tools against which we could compare. Given that we did not want to blindly test for differences in every possible LIWC category, we chose categories that had been shown to be important to narratives by some previous work, and this one was set of such categories that we decided to explore. While some of the results were not statistically significant, others, such as differences in the use of Cognition words and personal pronouns did differ significantly between narratives and non narratives in ways that help to confirm previous findings about narrative texts.
>
> 6) Based on reviewers' comments, in order to demonstrate the types of tweets that exist in our dataset, we conducted a manual thematic analysis of 400 tweets. 100 tweets were sampled from our full dataset from each of the following 4 categories: narrative information, non-narrative information, narrative misinformation, and non-narrative misinformation. Following a standard thematic analysis procedure, we first read through the sampled tweets, and then determine the main themes that existed. We coded each tweet and present the aggregate results below, including each theme, the percentage of the data subset that contained that theme, and one (paraphrased) example tweet to illustrate the theme. We will include these results in a formatted table in the final version of the paper, along with a discussion, if accepted.
>
> non-narrative information
>
> - **don't worry about what's in the vaccine (30%)** Example: "If you've ever been to that KFC on the street, don't worry about what's in the vaccine."
> - **get vaccinated (13%)** Example: "Will be getting vaccine next week. _/\_"
> - **other (10%)** "My sister argues with me about this vaccine everyday!"
> - **sharing news (9%)** Example: "The government will now begin vaccinating kids below the age of 10, starting Wednesday"
> - **questioning/debunking conspiracy theories (7%)** Example: "5G towers don't cause covid and there is no scientific evidence to support that claim. Stop spreading misinformation and please follow cdc guidelines and save lives"
> - **frustration towards those not following guidelines (6%)** Example: "So frustrating to see selfish people not follow rules. The pandemic isn't over yet. Please stay home and follow the rules."
> - **sharing local information (6%)** Example: "vaccine information: vaccinations available for ages 14+ this Sunday 3/09 at ABC Pharmacy in Carlton. If interested please call their number at 999-999-9999 #stay vaccinated"
> - **vaccine efficacy (6%)** Example: "The vaccine protects you and your loved ones from this deadly virus. Please believe in science and get vaccinated and save lives."
> - **mislabeled - should be narrative (4%)**
> - **public health tips (3%)** Example: "Stay indoors, wear masks and always sanitize your hands when you touch public areas."
> - **joke (3%)** Example: "my American friend says covid is a planned agenda, but what about Bill Gates acquiring large areas of farm lands?LOL"
> - **criticizing government (3%)** Example: "Only 6% of the adults have received both doses. This is bad governance at its best. You should learn from South Korea"
>
>
> narrative information
> - **got vaccinated and feeling positive (39%)** Example: "I got my second shot yesterday. I can finally go visit my mom and dad this weekend and i feel so grateful!!! <3"
> - **feeling/anticipating vaccine side effects (15%)** Example: "Got the jab two days ago and i am really feeling the effects. My arm is still sore and i feel the chills too. Not going to work tomorrow"
> - **friends/family member received the shot/feeling positive (11%)** Example: "My mom and dad got their first shot today and i feel so relived. Let's defeat this pandemic together. Peace"
> - **eager to receive vaccine (10%)** Example: "I could finally book my appointment for the vaccine. I am excited and can't wait to get the shot and meet my friends"
> - **someone got covid (3%)** Example: "Robert did not want to get vaccinated and guess what, he caught the virus and he's sick!!!!"
> - **vaccine scheduling (3%)** Example: "I have been trying to book vaccine appointment for weeks and still haven't made it, i feel so disappointed. Urghhhh!!!!!"
> - **friends/family member vaccine scheduled/feeling positive (3%)** Example: "My 78yo grandmother could finally book vaccine appointment for tomorrow. I feel so happy and grateful!! "
> - **expressing gratitude towards healthcare workers (3%)** Example: "i got my first shot yesterday and I thank all the frontline workers who have done phenomenally well during this pandemic. You saved countless lives. Feeling grateful!!"
> - **other (13%)**  include themes like "critisizing govt's handling of covid", "a joke", "setting example for others by getting vaccinated" etc
>
> non-narrative misinformation
>
> - **suspecting vaccine as gene therapy (19%)** Example: "Quick reminder, this isn't any normal vaccine. THIS IS GENE THERAPY, which will make a human forever slave to the big tech"
> - **vaccine hesitancy and concerns (19%)** Example: "They be calling them idiots for not wanting a rushed untested vaccine, when it's the vaccine that causes infertility"
> - **covid is a bioweapon (18%)** Example: "Wake up!! This is not a virus, it's a carefully launched bioweapon."
> - **govt/large corp (14%)** Example: "Fauci knows about the man-made virus long ago, it's all a cover up by the govt to keep the public under their shoes."
> - **covid is a depopulation agenda (13%)** Example: "There are 10 million unplanned pregnancy terminations since the beginning of this virus. Don't be stupid, this is a depopulation attempt by Bill Gates and the govts to control population growth."
> - **covid home remedy (10%)** Example: "Take mint leaves, honey, garlic and onion to prevent covid. you don't need the vaccine!!!!!!!!!"
> - **other (7%)** Example: "Sign this petition people! covid is hoax. #stopdepopulation #novax "
>
> narrative mis-information
> - **conspiracy theory (38%)** Example: "Fauci knew that the virus has been cooking in the lab since 2005 but he covered it up and now Bill Gates is inserting chips in our body through vaccine to track us"
> - **vaccine related deaths and side effects (28%)** Example: "A boy in Netherlands found dead 30 hours after taking his second covid shot. He was fit until a day before he took that damn vaccine!!!!"
> - **vaccine hesitancy (20%)** Example: "I will not take that vaccine until i get pregnant and give birth to a baby. I don't want to lose my fertility to a manufactured virus and supposed vaccine -_-"
> - **other (8%)** Example: "I unfriended a good friend of mine yesterday because she's totally into this  5G conspiracy theory shit. I don't have any regrets"
> - **covid home remedy (6%)** Example: "I have been eating garlic, one table spoon of honey, some onions before i go to bed and let me tell you, i have not had covid since! This totally works! and is the best solution for the virus."

---

### Official Review · Reviewer_uo6d · 2023-08-04

**Soundness:** 3

**Excitement:**

3: Ambivalent: It has merits (e.g., it reports state-of-the-art results, the idea is nice), but there are key weaknesses (e.g., it describes incremental work), and it can significantly benefit from another round of revision. However, I won't object to accepting it if my co-reviewers champion it.

**Missing References:**

Authors should explore the stance, morality, and reason detection task on COVID-19 by Pacheco et al 2022 (A holistic framework for analyzing the covid-19 vaccine debate).

Also for COVID-19 moral foundation and theme, authors could look into work by Islam and Goldwasser 2022 (Understanding COVID-19 Vaccine Campaign on Facebook using Minimal Supervision).

**Paper Topic And Main Contributions:**

This paper explores the relationship between narrative communication and health misinformation on social media, focusing on Twitter. The authors manually labeled a subset of tweets from misinformation datasets to identify narratives and then used the best model (their case RoBERTa) to extend these labels to the entire datasets. They found that narrativity can increase engagement with misinformation, especially for users with many followers.

**Reasons To Accept:**

The analysis part of the paper where the authors used statistical modeling to find out the relationship among narrativity, misinformation, follower count, and user engagement.

**Reasons To Reject:**

1. Line 327, 14.5k tweets were labeled by the annotators and the best RoBERTa model. If the annotators did 3k annotation,  It is not clear how the rest of the tweets are labeled by the model. Does it mean around 11.5k tweets were used as test data?

2. As data annotation was mentioned as one of the contributions, the authors could show some examples of annotated data and annotation procedures.

3. Authors could include a thematic analysis of the data.

**Reproducibility:**

2: Would be hard pressed to reproduce the results. The contribution depends on data that are simply not available outside the author's institution or consortium; not enough details are provided.

**Reviewer Confidence:**

4: Quite sure. I tried to check the important points carefully. It's unlikely, though conceivable, that I missed something that should affect my ratings.

---

> ### Author Rebuttal · Authors · 2023-08-29
>
> Thank you for taking the time to read and engage with our work! We appreciated your insights and suggestions. Below we have done our best to address all of your questions and concerns.
>
> 1) If the annotators did 3k annotation, It is not clear how the rest of the tweets are labeled by the model. Does it mean around 11.5k tweets were used as test data?
> Thanks for the question -- to clarify: the full dataset consists of 14561 tweets. We manually annotated a subset of 3000 tweets from the actual dataset using the process described in section 3.2, and then used a (80-10-10) split of that data to train our models as described in section 4. The best-performing model found in section 4 was then used to label the remaining 11.5k tweets for narrativity to use in the analyses in section 5.
> We would like to clarify that those 11.5k tweets are not considered part of the test set because our test set is data that has been labeled by human annotators in order to evaluate the performance of the model. Our test set is a representative sample of the full 14.5k that was selected in order to establish the generalization performance of our detection models, but these 11.5k were previous unlabeled data that we would like to analyze. Thanks for pointing this out; we will clarify this in the paper at the beginning of section 5 so it is completely clear how all of the portions of the data were labeled. Note that based on a question from R3, we also re-ran all of our experiments in section 5 using only the manually labeled 3k tweets to confirm that the results are in fact consistent, only lacking in the statistical power that we can achieve with the full set of 14.5k tweets.
>
> 2) Regarding examples of data annotation: the Annotation procedure is described in Section 3.2, with detailed guidelines in Appendix A for your reference, however we chose not show the exact examples of tweets based on prior scholarly works such as the work by Fiesle and Proferes: “Participant” perceptions of Twitter research ethics." Social Media+ Society 4.1 (2018): 2056305118763366. which suggests that users prefer not to have their exact messages presented in research papers or being used as examples. Our commitment to not directly showcasing examples stems from our ethical stance to safeguard the privacy of the authors of the tweets. However, based on your helpful suggestion, we conducted a thematic analysis, which summarizes the main themes that were present in the tweets so that you can understand the types of tweets that were annotated as narrative or non narrative, along with some paraphrased examples. The results will be added to the paper if accepted and we have included them at the end of this rebuttal (see item 5) for your reference.
>
> 3) We also noticed that we were given a low score for reproducibility (2) which contains the comment "Would be hard pressed to reproduce the results. The contribution depends on data that are simply not available outside the author's institution or consortium; not enough details are provided." We wanted to clarify that we will release our annotations and all data can be rehydrated using Twitter API; all of the annotation procedures and guidelines will be shared, pre-trained models will be shared, and other tools such as LIWC are available and will produce consistent results for anyone who uses them on our data. While we agree that the annotation process itself might have some variation depending on the annotators (as with most annotation procedures), we believe that the rest of our pipeline will be reproducible given the content of the paper and the aforementioned resources that we plan to release.
>
> 4) Thank you for pointing out these useful references which can be added to our paper.
>
> 5) Based on reviewers' comments, in order to demonstrate the types of tweets that exist in our dataset, we conducted a manual thematic analysis of 400 tweets. 100 tweets were sampled from our full dataset from each of the following 4 categories: narrative information, non-narrative information, narrative misinformation, and non-narrative misinformation. Following a standard thematic analysis procedure, we first read through the sampled tweets, and then determine the main themes that existed. We coded each tweet and present the aggregate results below, including each theme, the percentage of the data subset that contained that theme, and one (paraphrased) example tweet to illustrate the theme. We will include these results in a formatted table in the final version of the paper, along with a discussion, if accepted.
>
> non-narrative information
>
> - **don't worry about what's in the vaccine (30%)** Example: "If you've ever been to that KFC on the street, don't worry about what's in the vaccine."
> - **get vaccinated (13%)** Example: "Will be getting vaccine next week. _/\_"
> - **other (10%)** "My sister argues with me about this vaccine everyday!"
> - **sharing news (9%)** Example: "The government will now begin vaccinating kids below the age of 10, starting Wednesday"
> - **questioning/debunking conspiracy theories (7%)** Example: "5G towers don't cause covid and there is no scientific evidence to support that claim. Stop spreading misinformation and please follow cdc guidelines and save lives"
> - **frustration towards those not following guidelines (6%)** Example: "So frustrating to see selfish people not follow rules. The pandemic isn't over yet. Please stay home and follow the rules."
> - **sharing local information (6%)** Example: "vaccine information: vaccinations available for ages 14+ this Sunday 3/09 at ABC Pharmacy in Carlton. If interested please call their number at 999-999-9999 #stay vaccinated"
> - **vaccine efficacy (6%)** Example: "The vaccine protects you and your loved ones from this deadly virus. Please believe in science and get vaccinated and save lives."
> - **mislabeled - should be narrative (4%)**
> - **public health tips (3%)** Example: "Stay indoors, wear masks and always sanitize your hands when you touch public areas."
> - **joke (3%)** Example: "my American friend says covid is a planned agenda, but what about Bill Gates acquiring large areas of farm lands?LOL"
> - **criticizing government (3%)** Example: "Only 6% of the adults have received both doses. This is bad governance at its best. You should learn from South Korea"
>
>
> narrative information
> - **got vaccinated and feeling positive (39%)** Example: "I got my second shot yesterday. I can finally go visit my mom and dad this weekend and i feel so grateful!!! <3"
> - **feeling/anticipating vaccine side effects (15%)** Example: "Got the jab two days ago and i am really feeling the effects. My arm is still sore and i feel the chills too. Not going to work tomorrow"
> - **friends/family member received the shot/feeling positive (11%)** Example: "My mom and dad got their first shot today and i feel so relived. Let's defeat this pandemic together. Peace"
> - **eager to receive vaccine (10%)** Example: "I could finally book my appointment for the vaccine. I am excited and can't wait to get the shot and meet my friends"
> - **someone got covid (3%)** Example: "Robert did not want to get vaccinated and guess what, he caught the virus and he's sick!!!!"
> - **vaccine scheduling (3%)** Example: "I have been trying to book vaccine appointment for weeks and still haven't made it, i feel so disappointed. Urghhhh!!!!!"
> - **friends/family member vaccine scheduled/feeling positive (3%)** Example: "My 78yo grandmother could finally book vaccine appointment for tomorrow. I feel so happy and grateful!! "
> - **expressing gratitude towards healthcare workers (3%)** Example: "i got my first shot yesterday and I thank all the frontline workers who have done phenomenally well during this pandemic. You saved countless lives. Feeling grateful!!"
> - **other (13%)**  include themes like "critisizing govt's handling of covid", "a joke", "setting example for others by getting vaccinated" etc
>
> non-narrative misinformation
>
> - **suspecting vaccine as gene therapy (19%)** Example: "Quick reminder, this isn't any normal vaccine. THIS IS GENE THERAPY, which will make a human forever slave to the big tech"
> - **vaccine hesitancy and concerns (19%)** Example: "They be calling them idiots for not wanting a rushed untested vaccine, when it's the vaccine that causes infertility"
> - **covid is a bioweapon (18%)** Example: "Wake up!! This is not a virus, it's a carefully launched bioweapon."
> - **govt/large corp (14%)** Example: "Fauci knows about the man-made virus long ago, it's all a cover up by the govt to keep the public under their shoes."
> - **covid is a depopulation agenda (13%)** Example: "There are 10 million unplanned pregnancy terminations since the beginning of this virus. Don't be stupid, this is a depopulation attempt by Bill Gates and the govts to control population growth."
> - **covid home remedy (10%)** Example: "Take mint leaves, honey, garlic and onion to prevent covid. you don't need the vaccine!!!!!!!!!"
> - **other (7%)** Example: "Sign this petition people! covid is hoax. #stopdepopulation #novax "
>
> narrative mis-information
> - **conspiracy theory (38%)** Example: "Fauci knew that the virus has been cooking in the lab since 2005 but he covered it up and now Bill Gates is inserting chips in our body through vaccine to track us"
> - **vaccine related deaths and side effects (28%)** Example: "A boy in Netherlands found dead 30 hours after taking his second covid shot. He was fit until a day before he took that damn vaccine!!!!"
> - **vaccine hesitancy (20%)** Example: "I will not take that vaccine until i get pregnant and give birth to a baby. I don't want to lose my fertility to a manufactured virus and supposed vaccine -_-"
> - **other (8%)** Example: "I unfriended a good friend of mine yesterday because she's totally into this  5G conspiracy theory shit. I don't have any regrets"
> - **covid home remedy (6%)** Example: "I have been eating garlic, one table spoon of honey, some onions before i go to bed and let me tell you, i have not had covid since! This totally works! and is the best solution for the virus."

---

### Official Review · Reviewer_eVvg · 2023-08-08

**Soundness:** 4

**Excitement:**

4: Strong: This paper deepens the understanding of some phenomenon or lowers the barriers to an existing research direction.

**Paper Topic And Main Contributions:**

In this paper, the authors annotate an existing misinformation tweet dataset for the presence of narratives. The authors selected two existing medical datasets. They then followed the annotation process through an iterative process involving discussion and revision. They first annotated some samples, and then the rest of the samples were automatically annotated using machine learning techniques.

The conclusions that the authors made are:
1. narratives involve higher user engagement than misinformation
2. narratives may help increase engagement of misinformation

The key contribution of the work is the dataset and the follow-up conclusions that the paper makes.

**Questions For The Authors:**

Questions are mentioned above.

There is an assumption that the dataset would be available for free use later.

**Reasons To Accept:**

A new dataset that might help modelling user narratives online especially related to misinformation.
The studies that the paper presents are interesting and useful to understand the importance of user narratives.

**Reasons To Reject:**

One of the key concerns is the quality of the automatically annotated samples. How much control we have to maintain the quality of the automatically generated samples is something that needs to be mentioned in the paper. While the authors have used different machine learning models, these models have their own pros and cons.
Besides that, how do we know that the number of instances that have been manually annotated are enough for the computational model to learn reliably?

**Reproducibility:**

3: Could reproduce the results with some difficulty. The settings of parameters are underspecified or subjectively determined; the training/evaluation data are not widely available.

**Reviewer Confidence:**

3: Pretty sure, but there's a chance I missed something. Although I have a good feel for this area in general, I did not carefully check the paper's details, e.g., the math, experimental design, or novelty.

---

> ### Author Rebuttal · Authors · 2023-08-29
>
> Thank you for taking the time to read and engage with our work! We appreciated your insights and suggestions. Below we have done our best to address all of your questions and concerns.
>
> Q1a) Concerns regarding the quality of the automatically annotated samples.
> A) We manually annotated a large subset (n=3000) of the original dataset (n~=15000); trained large language models on the data across three runs, and chose the best-performing model to annotate the rest of the dataset automatically. Our F1 scores in Table 3 measure the quality of the automatically annotated samples for our held-out test set which was specifically sampled so that it would be representative of the entire (previously) unlabeled portion of the dataset. Given the reported metrics on the test set held fairly consistently across multiple runs, we can expect that the performance on the rest of the held-out data will be similarly strong.
>
> Q1b) How much control do we have to maintain the quality of the automatically generated samples?
> A) If you are referring the tweets themselves, we would like to clarify that we did not generate any of the samples. They are all real tweets that come from existing datasets already labeled by experts for the presence of misinformation. The generative models we employed in one of our experiments only output a binary narrativity class label (Yes or No) using the prompting approach described in section 4 of the paper.
>
> Q2) How do we know that the number of instances that have been manually annotated is enough for the computational model to learn reliably?
> A) While it's generally true that the more quality data we have, the better our models learn from them, we believe that the number of data points (n=3000) we chose to annotate was sufficient. The F1 score, our best-performing RoBERTa model (F1=0.924) indicates high levels of precision and recall in the binary classification task, which would be difficult to achieve without sufficient training data for the task.
> Other recent works on narrative analysis such as the ones listed below, successfuly trained narrative detection models with < 1/3 of the data we used for training.
>
> Narrative detection in online patient communities. Anne Dirkson, Suzan Verberne, Wessel Kraaij [European Conference on Information Retrieval 2019]
>
> Narrative Detection and Feature Analysis in Online Health communities. Achyutarama Ganti, Steven Wilson, Zexin Ma, Xinyan Zhao, Rong Ma [Workshop on Narrative Understanding 2022]

---

### Meta-Review · Area_Chair_9Yos · 2023-09-24

**Recommendation:** 3

**Metareview:**

This paper examines the interplay between narrative communication style and misinformative health content on user engagement on Twitter. Three reviewers, who reviewed the submission, highlighted the strengths of the paper. R1 appreciated its valuable dataset, R2 appreciated the analyses, and R3 appreciated the paper’s outcome in motivating this line of research. However, all the reviewers raised significant concerns about the paper, some of which were addressed during the rebuttal phase by the authors—and the reviewers accordingly updated their ratings. Particularly the reviewers commended the authors on the additional thematic analysis.

---

### Decision · Program_Chairs · 2023-10-07

**Decision:**

Accept-Findings

**Comment:**

This paper examines the interplay between narrative communication style and misinformative health content on user engagement on Twitter. Three reviewers, who reviewed the submission, highlighted the strengths of the paper. R1 appreciated its valuable dataset, R2 appreciated the analyses, and R3 appreciated the paper’s outcome in motivating this line of research. However, all the reviewers raised significant concerns about the paper, some of which were addressed during the rebuttal phase by the authors—and the reviewers accordingly updated their ratings. Particularly the reviewers commended the authors on the additional thematic analysis.